



# Teleconnections and relationship between ENSO and SAM in reconstructions and models over the past millennium

Christoph Dätwyler[1], Martin Grosjean[1], Nathan J. Steiger[2], Raphael Neukom[1]

[1]Institute of Geography and Oeschger Centre for Climate Change Research, University of Bern, Bern, 3012, Switzerland
[2]Lamont-Doherty Earth Observatory, Columbia University, Palisades, NY, USA

*Correspondence to*: Christoph Dätwyler (christoph.daetwyler@giub.unibe.ch)

**Abstract.** The climate of the Southern Hemisphere (SH) is strongly influenced by variations in the El Niño-Southern Oscillation (ENSO) and the Southern Annular Mode (SAM). Due to the temporally very limited instrumental records in most parts of the SH, very little is known about the relationship between these two key modes of variability and its stability over

time. Here, we use proxy-based reconstructions and climate model simulations to quantify changes in tropical-extratropical SH teleconnections as represented by the correlation between the ENSO and SAM indices. Reconstructions indicate mostly negative correlations back to around 1400 CE confirming the pattern seen in the instrumental record over the last few decades. An ensemble of last millennium simulations of the model CESM1 confirms this pattern with very stable ensemble mean correlations around -0.3. Individual forced simulations, the pre-industrial control run and the proxy-based reconstructions

indicate intermittent periods of positive correlations and particularly strong negative correlations. The fluctuations of the ENSO-SAM correlations are not significantly related to solar nor volcanic forcing in both proxy and model data, indicating that they are driven by internal variability in the climate system.

Pseudoproxy experiments indicate that the currently available proxy records are able to reproduce the tropical-extratropical teleconnection patterns back to around 1600 CE. We analyse the spatial temperature and sea level pressure patterns during

periods of positive and particularly strong negative teleconnections in the CESM model. Results indicate no consistent pattern during periods where the ENSO-SAM teleconnection changes its sign. However, periods of very strong negative SH teleconnections are associated with negative temperature anomalies across large fractions of the extra-tropical Pacific and a strengthening of the Aleutian Low.

## 1 Introduction

The El-Niño Southern Oscillation (ENSO) and the Southern Annular Mode (SAM) are the Earth's and Southern Hemisphere's (SH) leading modes of interannual climate variability (Marshall, 2003; McPhaden et al., 2006), respectively. Although ENSO is a mode of tropical variability, it affects regional temperature and hydroclimate far into the mid-latitudes (Kiladis and Diaz, 1989; Cole and Cook, 1998; Moron and Ward, 1998; McCabe and Dettinger, 1999; Gouirand and Moron, 2003; Mariotti et al., 2005; Brönnimann et al., 2007) and some studies have even identified its influence in the Antarctic domain (Vance et al.,





2012; Roberts et al., 2015; Jones et al., 2016; Mayewski et al., 2017). SAM is the dominant mode of SH high-latitude
atmospheric variability, but its fluctuations affect climate also northwards into the sub-tropics, mainly by changing
precipitation patterns (Silvestri and Vera, 2003; Gillett et al., 2006; Hendon et al., 2007; Risbey et al., 2009). Despite the
prominent role of these two modes in driving regional SH climate, little is known about the interplay between them. This
interplay and its stability over time is, however, a key factor for understanding tropical-extratropical teleconnections in the
SH. Understanding SH climate dynamics and identifying the key drivers of interannual to multi-decadal variability requires
long-term quantifications of these dominant modes of variability.

The evolution of the tropical to extratropical teleconnections between ENSO and SAM prior to the mid-20th century is unclear.
The main cause for this knowledge gap is the low spatio-temporal coverage of instrumental observations across vast parts of
the SH. Reliable and consistent instrumental quantifications of the SAM only extend back to the start of the satellite era in
1979 (Ho et al., 2012). Station-based indices extend back to 1957. In the instrumental indices during the most recent decades
we see a strong negative correlation between the commonly used indices representing these two modes of climate variability
(L'Heureux and Thompson, 2006; Wang and Cai, 2013; Kim et al., 2017). A relationship between ENSO and SAM has been
found in various studies and with focus on different geographic regions (Ribera and Mann, 2003; Silvestri and Vera, 2003;
Carvalho et al., 2005; Fogt and Bromwich, 2006; L'Heureux and Thompson, 2006; Cai et al., 2010; Gong et al., 2010; Pohl et
al., 2010; Fogt et al., 2011; Clem and Fogt, 2013; Lim et al., 2013; Yu et al., 2015; Kim et al., 2017). However, they mainly
rely on reanalysis data and hence only consider (parts of) the instrumental period and satellite era. Many of these studies also
find a negative relationship between ENSO and SAM as seen in the instrumental indices over the last 50 years (Carvalho et
al., 2005; L'Heureux and Thompson, 2006; Cai et al., 2010; Gong et al., 2010; Pohl et al., 2010; Fogt et al., 2011; Ding et al.,
2012; Wang and Cai, 2013; Kim et al., 2017). But also different ENSO-SAM teleconnections and changes thereof at various
time periods have been detected (Fogt and Bromwich, 2006; Clem and Fogt, 2013; Yu et al., 2015).

While there is some literature about the stability of the individual ENSO and SAM indices over the past centuries (e.g. Wilson
et al., 2010; Villalba et al., 2012; Dätwyler et al., 2018; Dätwyler et al., 2019) and their driving factors (e.g., Dong et al., 2018),
hardly any studies exist that use palaeoclimate evidence to investigate the relationship between these modes of climate
variability back in time. A study based on palaeoclimate records by Gomez et al. (2011) found a waxing and waning
relationship between ENSO and SAM on century to millennial long time scales. Abram et al. (2014) found a significant
negative correlation between their SAM reconstruction and the SST reconstruction in the Niño3.4 region of Emile-Geay et al.
(2013) since 1150 CE. However, to our knowledge, there are no studies based on annually resolved palaeoclimate data that
assess the relationship between ENSO and SAM over the last millennium on sub-centennial time scales. Hence, still little is
known about the tropical to extratropical teleconnections established by ENSO and SAM over this time period. Also, it is not
known if the interplay between these indices exhibits long-term fluctuations and if these are driven by external forcing or
internal variability, for instance via multi-decadal modes of variability such as the Interdecadal Pacific Oscillation (IPO,
Folland et al., 1999; Power et al., 1999; Henley, 2017).



Climate models can help to identify driving factors of climatic variability and teleconnection changes. In a model environment, climate indices can be analysed over much longer periods than is possible with reanalysis data or using only instrumental measurements. Palaeoclimate reconstructions can then serve as a benchmark against which such analyses can be tested. Climate models rely on physically self-consistent processes and generate data that spatially cover the entire Earth's surface. These two characteristics provide the basis for pseudo-proxy experiments (PPE; Mann and Rutherford, 2002; Smerdon, 2012). Pseudo-proxies are the virtual equivalents of real-world proxy records in climate models. The reliability and robustness of reconstructions that are obtained using real proxy records can be evaluated by comparing them to reconstructions based on pseudo-proxies. In addition, pseudo- proxy based reconstructions have the advantage that they can be compared to the simulated climate, the "model truth", over the entire reconstruction period, not only the short overlap period with instrumental data as in the real-world situation.

With this study we aim to put the observed relationship between ENSO and SAM over the most recent decades into a long-term context. We investigate how well proxy-based ENSO and SAM reconstructions can reproduce the ENSO-SAM correlation pattern. These goals are achieved by combining the knowledge on the two climate modes over the instrumental period with palaeoclimate evidence from various proxy archives and complementing this information with knowledge of climate model runs over the past millennium. In a first step, we analyse the ENSO-SAM relationship with millennial-long reconstructions that are based on real-world proxy records. In a second step, this relationship is tested with model-based indices, which in addition helps disentangling possible effects of volcanic or solar forcing from internally driven variability. Thirdly, we carry out PPE which serve to test if reconstruction methods, decreasing data availability and quality back in time (i.e. proxy-inherent noise) influence or bias the reconstructed relationship between the two climate modes. Spatial patterns of climate during periods of particularly strong negative or reversed positive ENSO-SAM relationship are analysed in the model world to identify potential driving factors of SH teleconnection changes.

## 2. Data and Methods

### 2.1 Data

Our model analyses are based on the Community Earth System Model-Last Millennium Ensemble (CESM-LME; Otto-Bliesner et al., 2016) from which we use the sea surface temperature (SST) and pressure at sea level (PSL) variables. The ensemble consists of 13 fully forced runs covering the years 850-2005 and one pre-industrial control run of the same length. For our analyses we only use the data from 1000 to 2005. The volcanic forcing in the CESM model is adopted from Gao et al. (2008) and the solar forcing uses Vieira et al. (2011). The values of the forcings are nominally adjusted forcings (W/m$^2$) at the top of the atmosphere (TOA). The baseline for the total solar irradiance (TSI) is the Physikalisch-Meteorologisches Observatorium Davos (PMOD) composite over 1976-2006 (details see Schmidt et al., 2011).





As an instrumental index for ENSO, we use the NINO3.4 index based on the Extended Reconstruction Sea Surface
Temperature Version 4 (ERSSTv4) instrumental data set (Huang et al., 2015). It is defined as the area average SST anomaly
from 5°N-5°S and 170°-120°W (Barnston et al., 1997).

For SAM, the Marshall (Marshall, 2003) and Fogt (Fogt et al., 2009; accessible through
http://polarmet.osu.edu/ACD/sam/sam_recon.html) indices are used. To calculate the model SAM index we use the definition
of Gong and Wang (1999) who define it as the normalised zonal mean SLP difference between 40°-65°S.

## 2.2 Methods

We consider data quality to be sufficiently good post-1900 and hence use 31-year running correlations between the
instrumental indices back to 1915 (middle-year, i.e. corresponding to the first 31-year period ranging from 1900 to 1930).
Additionally, the post-1900 period covers the calibration intervals that were used for the ENSO and SAM reconstructions
(1930-1990 and 1905-2005).

### 2.2.1 Reconstructions

This study uses the newest available ENSO and SAM reconstructions. For the SAM this is the austral summer season SAM
reconstruction of Dätwyler et al. (2018). It uses the proxy records from Villalba et al. (2012), Abram et al. (2014), Neukom et
al. (2014), PAGES 2k Consortium (2017) and Stenni et al. (2017) and is based on the nested ensemble-based composite-plus-
scaling (CPS) method (Neukom et al., 2014).

In the case of ENSO, the palaeoclimate reconstruction is taken from Dätwyler et al. (2019), but adapted to the austral summer
season December to February (DJF, Fig. S1). That is, the sub-annually resolved proxy records (corals) were averaged over the
three-month interval DJF. The proxy data combine the PAGES 2k global temperature database (PAGES 2k Consortium, 2017),
the SH proxy records from Neukom et al. (2014), the Antarctic ice core isotopes from Stenni et al. (2017), and also ENSO-
related hydroclimate records from the northern hemisphere (Emile-Geay et al., 2013; Henke et al., 2017). In brief, the
reconstruction method is based on PCA: ENSO is quantified as the first PC of the available proxy records in each overlapping
80-year period over the last millennium (Dätwyler et al., 2019). The key advantage of this approach is that it does not require
formal calibration with instrumental data and is, therefore, not prone to the assumption that the relationships within the proxy-
instrumental data matrix are stable over time. The running approach allows us to capture changes in the strength of each proxy
record's contribution to the first PC over time. Such changes may happen when the relationship between the proxy and target
variable changes. For details we refer to the original paper (Dätwyler et al., 2019).

Although both reconstructions share some input proxy data, they are mostly independent (for details see SM Sect. 2).

### 2.2.2 Superposed Epoch Analysis

Superposed epoch analysis (SEA; Haurwitz and Brier, 1981; Bradley et al., 1987; Sear et al., 1987; McGraw et al., 2016) is
used to analyse whether a response to large volcanic eruptions can be detected in a time series. For comparisons with the



palaeoclimate reconstructions we use the eVolv2k volcanic forcing based on Toohey and Sigl (2017) with a threshold for the aerosol optical depth (AOD) of >0.15, which selects the 13 largest eruptions in the last millennium (1109, 1171, 1230, 1257, 1286, 1345, 1458, 1600, 1641, 1695, 1784, 1809 and 1815). Model data are compared to the volcanic forcing dataset used to drive the simulations (Gao et al., 2008). We consider the 13 strongest volcanic events in the last millennium with a forcing stronger than 2.5 W/m$^2$ (1177, 1214, 1259, 1276, 1285, 1342, 1453, 1601, 1642, 1763, 1810, 1816 and 1836). For

methodological details on the SEA see Dätwyler et al. (2019).

### 2.2.3 Pseudoproxy Experiments

For both ENSO and SAM we calculate reconstructions based on virtual proxy records in the model world to evaluate the "real-world" reconstructions described above in Sect. 2.2.1. For all real-word proxy records that contributed to the respective reconstructions their model-world equivalents are generated at the same geographic locations. The same reconstruction

methods as were used for the real-world proxy based reconstructions are then applied using the model ENSO and SAM indices as reconstruction targets.

We use two categories of pseudo-proxies. That is, perfect pseudo-proxies that correspond directly to the model climate variable (temperature or precipitation) each proxy record reflects (details see SM Sect. 2) and pseudo-proxies on which noise with a certain amount and structure is imposed. Depending on the record's archive type, two different types of pseudo-proxy records

are generated. Following the methodology in Steiger and Smerdon (2017), proxy-system-model (PSM) pseudo-proxies are used in the case of tree-ring width using the VS-Lite model (Tolwinski-Ward et al., 2011) based on monthly temperature and precipitation as input variables. $\delta^{18}$O coral records are based on a PSM with annual SST and sea surface salinity within the CESM1 simulations as input (Thompson et al., 2011). PSMs are designed to take into account specific physical and biological properties of their proxy archive. The resulting pseudo-proxies have a signal-to-noise ratio (SNR) of roughly 0.5 (Neukom et

al., 2018). For other archives (e.g., ice cores, lake and marine sediments, speleothems and historical documents) we generate statistical noise-based pseudo-proxies as described in Neukom et al. (2018). We use a signal to noise ratio of 0.5 by standard deviation, which is usually considered realistic for annually resolved proxy data (Wang et al., 2014), but rather conservative given the reduced correlation of local to large-scale climate in the CESM model (Neukom et al., 2018).

### 2.2.4 Spatial temperature and SLP patterns in models

Model temperature and SLP fields are analysed in order to identify possible spatial patterns in the climate system during periods of positive and particularly strong negative ENSO-SAM relationships. First, all years within the pre-industrial (1000-1850) period, where the ENSO-SAM correlations are reversed (i.e. positive) or strongly negative are selected in the 13 members of the CESM full-forced ensemble and in the pre-industrial control simulation. As threshold for reversed teleconnections (positive ENSO-SAM correlations), we use a value of 0.26, which is three standard deviations above the mean

correlation across all datasets (-0.30). Within all 11914 model years, this value is exceeded during 53 years (centre years of the 31-year running correlations). For strong teleconnections, we use a threshold of -0.67, which is two standard deviations





below the average correlation. This threshold is exceeded during 281 years in total. Note that our conclusions are robust to using different thresholds (Figs. S4-S5). Note that the years exceeding these thresholds are usually not isolated but distributed over 7 (31) blocks with an average length of 7.6 years (9.1 years) for reversed (strong) teleconnections. We then calculate the

average model temperature and SLP at each grid cell over the globe during the selected years. Significance is tested similarly to the SEA, by generating 1000 averages of randomly chosen years using the same amount and length of blocks as described above. If the anomalies are outside the 95% range of the random averages, they are considered significantly higher or lower than expected by chance.

## 3 Results and Discussion

### 3.1 ENSO-SAM relationship in real-world data


The correlations between the instrumental ENSO and SAM indices are largely negative (Fig. 1), confirming results from the literature (Carvalho et al., 2005; L'Heureux and Thompson, 2006; Cai et al., 2010; Gong et al., 2010; Pohl et al., 2010; Fogt et al., 2011; Ding et al., 2012; Wang and Cai, 2013; Kim et al., 2017). Particularly during the period of high data quality back to the 1970s, the relationship is strongly negative. In the longer and less certain instrumental dataset (Fogt SAM index), a

breakdown of the SAM-ENSO relationship is visible around 1955.

Running correlations between the real-world proxy-based austral summer ENSO and SAM reconstructions show an extension of this mostly negative relationship back to the middle of the 14[th] century, with the exception of a short period around 1700, where the sign of the relationship is reversed, i.e. positive (Fig. 1). Between 1150 and 1350 we see a period of mostly positive correlations, whereas during the earliest 150 years (1000-1150) generally negative correlations prevail. Over the whole

millennium a trend towards more negative correlations, peaking in the 19[th] and 20[th] century, can be observed. The SEA assessing the impact of volcanic eruptions on the ENSO-SAM correlations does not yield a significant result (Fig. S2). In addition, we do not see any significant correlation between the ENSO-SAM relationship and solar forcing (not shown), indicating that fluctuations in this relationship are largely internally driven.

### 3.2 ENSO-SAM relationship in the CESM model

To test whether the breakdowns in the reconstructed ENSO-SAM correlations back in time and the lack of response to external forcing are realistic features or an artefact of decreasing proxy data availability and quality, we now compare the results with correlations from model simulations.

The mean of the running correlations across the 13 forced simulations confirms the negative correlations in the 20[th] century seen in the observational datasets, indicating consistent performance of the model in simulating the ENSO-SAM relationship

over this period. The ensemble mean correlations remain negative for the entire millennium with only relatively little variation (Fig. 2). In addition to this consistent average negative response, the individual 13 running correlations for each simulation also show periods with breakdowns in the negative correlation and intermittent changes to positive relationships (Fig. S3).





However, in most of the 13 model runs, these changes are of lower magnitude compared to the palaeoproxy reconstructions. Similar to the observational data, the SEA of the 31-year running correlations in the individual simulations do not yield a

consistent response of the ENSO-SAM relationship to volcanic eruptions (SM Sect. 3). There is no significant relationship between the mean running correlations of the model ENSO and SAM indices and the solar forcing (not shown). Since neither solar nor volcanic forcing appear to be driving the fluctuations in the ENSO-SAM correlations, they are likely to be caused by internal variability. This is confirmed by the pre-industrial control run, which also shows the average negative correlations, with superimposed fluctuations at multi-decadal time-scales (Fig. 2, Fig. S3).

**3.3 Pseudo-proxy based ENSO and SAM reconstructions**

The general pattern of a negative correlation between ENSO and SAM and no significant influence of external forcing on this relationship is consistent among real-world proxy-based reconstructions and models, but the reconstructions show a weakening of the correlations back in time, which is not evident in the model data. To test whether the reconstructed pattern may be an artefact, we now compare our results to the pseudoproxy experiments.

By taking the mean of all 13 running correlations between the perfect pseudo-proxy based ENSO and SAM reconstructions we can observe that the running correlations stay negative over the whole millennium with a slight trend towards more negative correlations towards present (Fig. 3a). That is, we find a similar mean relationship as can be observed with the model index reconstructions, with the only difference of a slight weakening in the early most period. This similar mean relationship also indicates that the locations and number of proxy records are suitable to capture the ENSO and SAM signals.

To quantify the influence of noise in the proxy records, we perform additional PPE using noisy pseudoproxies with realistic correlations to local climate. The resulting relationship shows strong negative correlations (around -0.6) only in the 19[th] and 20[th] century (Fig. 3b). The noise in the pseudo-proxies results in weaker negative correlations pre-1600, when the number of proxy records contributing to the reconstructions decreases notably. A significant decrease in ENSO reconstruction skill prior to 1600 due to proxy availability and quality has also been seen in Dätwyler et al. (2019). The signal seen in the instrumental

data over the last decades (stable negative ENSO-SAM correlations) is thus lost only partly in the perfect proxy experiment and the loss takes place gradually over the entire millennium. In contrast, the noisy PPE suggests that the signal decreases rapidly prior to 1800 and disappears in the pre-1600 period. This result suggests that the decreasing strength in the ENSO-SAM relationship seen in the early phase of the real-world proxy-based reconstructions may be an artefact of noise inherent in the proxy data.

The running correlations in Figs. 3c and 3d show the skill of the real-world and pseudo-proxy ENSO and SAM reconstructions, respectively. Note that the correlations between the reconstructions and the target can only be quantified over the instrumental period for the real-world proxy and observational data, but over the entire millennium for the PPE. In the case of perfect pseudo-proxies for both ENSO and SAM, the mean running correlations between the model index and the pseudo-proxy reconstruction remain highly positive (~0.7-0.9) over the whole analysed period with lowest values for the SAM (~0.5-0.6) in

the 11[th] and 12[th] century. In case of noisy pseudo-proxies for the ENSO, the correlations break down significantly prior to 1800





when they remain positive but on a lower level. In the case of SAM, the noisy pseudo-proxies do not result in an equally strong breakdown of the running correlations. Correlations are about 0.2 lower for the noisy PPE throughout most of the reconstruction period; only around 1300 they start to break down to levels around 0.25 between 1000-1200. This indicates that the SAM proxy network is more stable and representative even with fewer predictors, at least in the CESM1 model world. The

performance of the real-world post-1900 reconstructions (red and green lines in Figs. 3c and 3d) is closer to the perfect pseudo-proxy reconstructions than to the noisy PPE. This implies that SNR = 0.5 is likely to be too conservative and the case of perfect pseudo-proxies may be more representative for the real-world. This is consistent with Neukom et al. (2018), who based on the same climate model, also showed that perfect pseudo-proxies can yield results closer to those obtained using real-world proxy records than if noisy proxies with a too high level of noise are used. A possible reason for this observation is the fact that local

climate is less correlated to large-scale indices in CESM1, compared to instrumental data (Neukom et al., 2018).

Since the relationship between ENSO and SAM using perfect pseudo-proxy records resembles more the relationship seen in the model truth than when using noisy pseudo-proxies, we argue that proxy-based reconstructions can reproduce the ENSO-SAM correlation pattern in a realistic way, if there are enough proxy records contributing to the reconstructions. Even with a conservative choice in the amount of added noise to the pseudo-proxy records, a pronounced negative correlation between

ENSO and SAM is observable, which further speaks for a negative relationship between ENSO and SAM. The rapid breakdown of the signal in the noisy PPE prior to 1800 due to proxy noise may be over-pessimistic and the gentler decrease in the perfect PPE a more realistic representation of the real-world situation. As such, the low ENSO-SAM correlations seen in the real-world proxy-based reconstructions (Fig. 1) in the earliest few centuries of the reconstruction period are likely to be results of the lower number and quality of proxy records.

In contrast, the temporal fluctuations of the correlations in the real-world reconstructions in the second half of the millennium may be real and not influenced by proxy data availability and quality. Since there was also no change in the number of proxy records contributing to the ENSO reconstruction between 1600 and 1800 and only very few dropping out in the SAM reconstruction between 1600 and 1850, a loss of key proxy records is unlikely to be responsible for e.g. the positive swing in the ENSO-SAM relationship around 1700 or the strong negative correlations around 1765. Hence, the strong deviations

towards positive correlations as seen around 1700 and the particularly strong negative correlations around 1570, 1765 and in the mid-19[th] century are possibly generated by internally driven teleconnection changes.

### 3.4 Spatial temperature and SLP patterns in the model world during periods of positive and very strong negative teleconnections

Given we find no significant influence of external forcing on fluctuations in the ENSO-SAM teleconnections (Sect. 3.1 and

3.2), changes in this relationship are likely internally driven. Within the last few centuries we have found periods in the real-world proxy-based reconstructions during which the sign of the ENSO-SAM correlation changes to positive values. Other periods were identified where the relationship is particularly strongly negative. Are there any distinct spatial patterns in the climate system that lead to such positive or strong negative correlations? To identify such patterns, we analyse similar



anomalies in the model data as described in the methods section. The resulting temperature and SLP patterns during years of
positive (top) and strong negative (bottom) teleconnections are shown in Fig. 4, significant anomalies are indicated with black
stippling.

Temperature patterns during positive teleconnections resemble the spatial patterns of a negative Interdecadal Pacific
Oscillation (Henley, 2017). The average Tripole Index (TPI) is indeed negative during these years but not significantly lower
than during "normal" years (Fig. 4, second column). The SLP pattern resembles a negative PNA (Pacific-North American
pattern; Wallace and Gutzler, 1981) and positive NAO (North Atlantic Oscillation; Walker and Bliss, 1932) phase, however
only very limited areas exhibit significant anomalies. For example, the Aleutian Low exhibits positive but non-significant
anomalies (Fig. 4, top right).

During periods of very strong negative ENSO-SAM correlations, most of the globe exhibits colder than normal temperatures
with significant anomalies over the north-eastern and southern Pacific. This spatial patterns now resembles a positive IPO
fingerprint, however the positive TPI anomalies are, again, not significant. The SLP pattern during strong teleconnections
shows very strong negative anomalies in the northernmost Pacific resulting in a significantly negative Aleutian Low. Figure 4
also suggests significant negative anomalies in the south-eastern Pacific, thus a weakening of the South Pacific Anticyclone.
However, this result is not robust to parameter changes (Figs. S4 and S5).

The combination of a strong Aleutian Low and a positive IPO is consistent with literature (e.g., Newman et al., 2016), and
effects of the IPO on ENSO teleconnections and related interplay with Northern Hemisphere pressure patterns have also been
described (Dong et al., 2018, and references therein).

Dynamical links between ENSO, SAM and NH dynamics have already been reported in the literature. E.g., Marini et al. (2010)
describe a link between SAM and the Atlantic Meridional Overturning Circulation (AMOC) on different time scales. And Li
et al. (2015) found that after the late 1990s the co-variability of the Aleutian Low and the negative SAM phase are linked by
changes in the ENSO cycle. Our analysis suggests that there is no single consistent temperature or SLP pattern that explains
the internally driven reverse of teleconnections in the CESM simulations. In contrast, periods of strong tropical-extratropical
teleconnections such as reconstructed in the mid-19[th], mid-18[th] and mid-16[th] centuries may have been related to a strengthening
of the Aleutian low and associated cooling in extra-tropical Pacific SSTs. A test of this hypothesis has to await upcoming high
quality climate field reconstructions of temperature and SLP and associated climate indices.

## 5 Conclusions

We present, for the first time, a multi-century analysis of SH tropical-extratropical teleconnections in palaeoproxy observations
and model data. CESM1 climate model simulations confirm what we see in the instrumental indices and in proxy-based ENSO
and SAM reconstructions: A generally negative correlation between the ENSO and SAM with a multi-century average around
-0.3. This relationship is perturbed only by internal variability as we do not find any significant response to solar and volcanic
external forcing. The relationship between ENSO and SAM as seen in the model indices agrees well with the relationship





established by perfect pseudo-proxy record reconstructions indicating that reconstruction methods and geographic distribution of the proxy records are well suited to capture the essential ENSO and SAM signal. If a significant amount of noise is added to the pseudo-proxy records, the negative correlations weaken pre-1800 and even further pre-1600, when the number of available proxy records drops below a critical threshold (~30 proxy records) indicating reduced reliability of the reconstructed

ENSO-SAM correlations during this period.

Reconstructions and model simulations indicate that there are repeated periods where the tropical-extratropical teleconnections break apart or are particularly strong. We find no consistent spatial climate patterns during periods of reversed ENSO-SAM correlations. In contrast, periods of particularly strong negative correlations are associated with a strong Aleutian Low and negative temperature anomalies across the extra-tropical Pacific.

Our results may serve as a basis for future research further studying the long-term behaviour and stability of teleconnections in the Southern Hemisphere. This will help placing observed and predicted SH climate changes, such as a potential expansion of the Hadley Cell (e.g., Hu et al., 2013) or changes in Antarctic sea-ice extent (e.g., Turner et al., 2009; Parkinson and Cavalieri, 2012) into a larger spatio-temporal context by assessing their relationship to hemispheric-scale teleconnection changes.


**Data availability**

[Link to NOAA paleoclimatology database].

**Supplementary material**

[Link to supplementary material]

**Author contributions**

RN and CD designed the study. Data analysis was led by CD with contributions from RN. The writing was led by CD with exception of Chapter 3.4 and the abstract, which were mainly written by RN. All figures were made by CD, except Fig. 4,

which was made by RN. Pseudo-proxy data were provided by NJS and RN. All authors jointly discussed and contributed to the writing.

**Competing interests**

The authors declare that they have not conflict of interest.


**Acknowledgements**

This work partly resulted from contributions to the Past Global Changes (PAGES) 2k initiative. Members of the PAGES2k consortium are thanked for providing public access to proxy data and metadata. This study was supported by the Swiss National




Science Foundation (SNF) Ambizione grant PZ00P2_154802 and by the United States National Science Foundation grant
NSF-AGS 1805490.

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



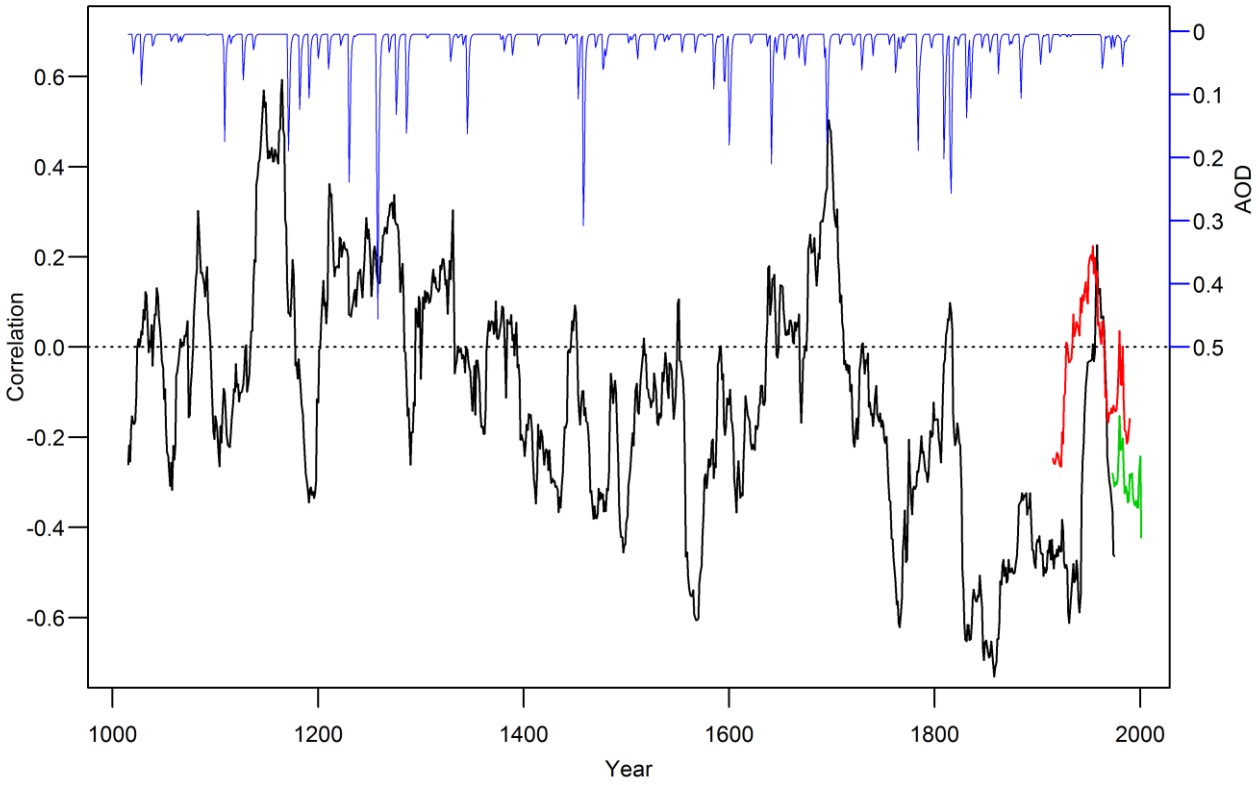

**Figure 1: 31-year running correlations between real-world DJF ENSO and SAM reconstructions (black) together with 31-year running correlations of the instrumental indices (red: ENSO and Fogt SAM index, green: ENSO and Marshall SAM index). The volcanic forcing (Toohey and Sigl, 2017) is shown in blue. Years on the x-axes reflect the middle year of the 31-year running correlations.**






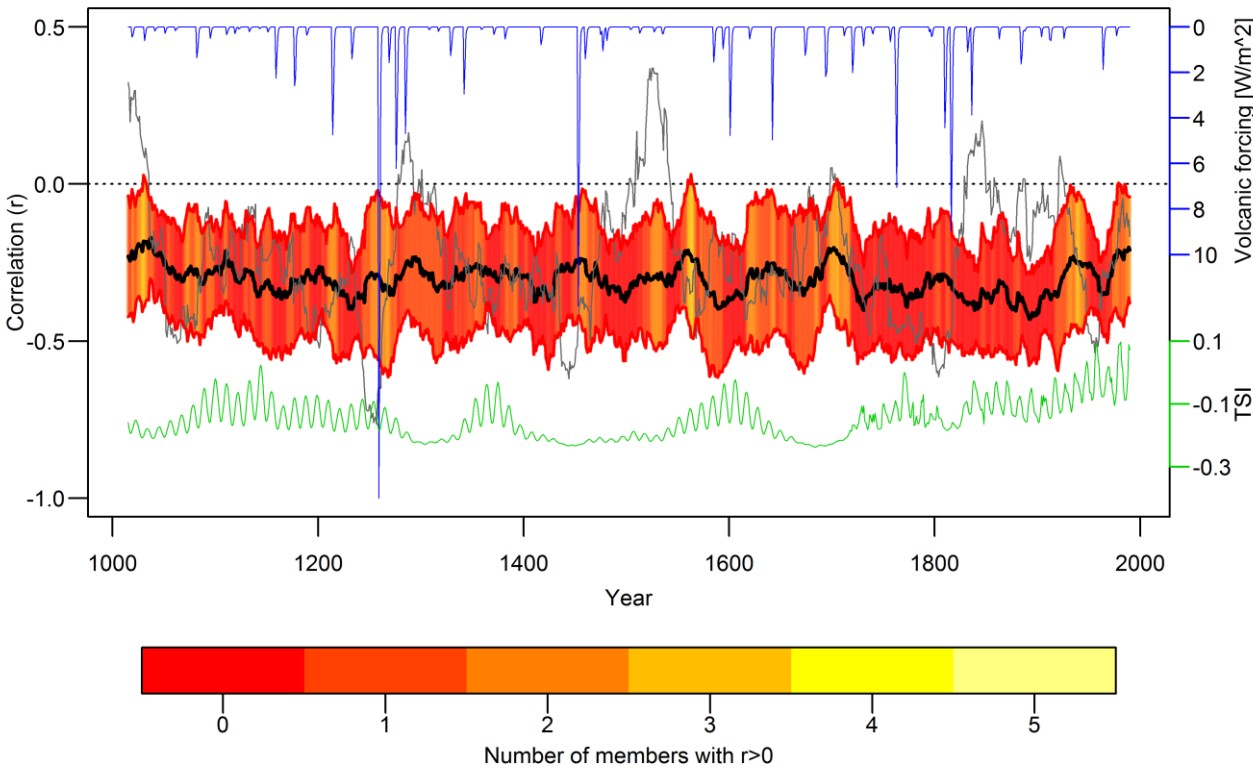

**Figure 2: Mean of the 31-year running correlations between model ENSO and SAM indices (models 1-13, thick black line) and +/-
1 standard deviation (red lines). The grey line shows the running correlations between ENSO and SAM for the control run. The
volcanic forcing is shown in blue and the solar forcing in green. The red to yellow shading indicates for each year the number of
510   running correlations among the individual 13 models that show a positive (reversed) sign.**





**Figure 3:** Panel (a) shows the mean of the 31-year running correlations over all 13 models between the perfect pseudo-proxy based ENSO and SAM reconstructions. Panel (b) shows the same as panel (a) but using the noisy pseudo-proxy based ENSO and SAM reconstructions. The grey lines correspond to +/- 1 standard deviation. Panel (c) displays in black (blue) the mean 31-year running correlations over the 13 models between the model ENSO index and the perfect (noisy) pseudo-proxy based ENSO reconstructions. The red line shows the 31-year running correlations between the real-world proxy records based ENSO reconstruction and the instrumental ENSO index. Panel (d) is the analogue of panel (c) but for the SAM. The red (green) line displays the 31-year running correlations between the real-world proxy records based SAM reconstruction and the instrumental Fogt (Marshall) SAM index. Panels (e) and (f) are enlargements of panels (c) and (d) over the 20th century. The red to yellow shading in panels (a) and (b) indicates the number running correlations among the individual 13 models that show a positive (reversed) sign.



**Figure 4 Simulated climate anomalies during years of positive (top) and particularly strong negative (bottom) ENSO-SAM correlations. Left: Surface temperature anomalies. Second column: IPO anomalies. Third column: SLP anomalies. Right: Aleutian low anomalies. Black stippling represents significant results in maps. Bold lines in the boxplots represent the median of random-sampled anomalies, boxes the interquartile range and whiskers the 95% range. Green circles represent the average anomalies during years of reversed or strong teleconnections**