# Peer review of "Teleconnections and relationship between ENSO and SAM in reconstructions and models over the past millennium"

_Climate of the Past, 2019_

## Referee Comment (RC1) · Anonymous Referee #1 · 22 Oct 2019

Overview – the authors are examining relationships between austral summer ENSO and SAM indices from reconstructions and model simulations over the last millennia. As these relationships are important for understanding climate variations across the mid and high latitudes of the Southern Hemisphere, and no research to my knowledge has been done on these relationships this far back in time, the work is very important.

The paper is extremely well written, concise, and easy to follow. I enjoyed it very much, and suggest it be published after the authors address my three main concerns, and two very minor suggestions.

Major concerns:

[Figure]

1. Figure 1 – is there a way that uncertainty in the reconstructions (both the Fogt and the proxy-based) can be accounted for when calculating the correlation, and this can be shown as some sort of confidence interval around the correlations? While you can do this as a statistical test that the correlation is zero (95% confidence interval on the correlation magnitude itself), I think it would be more telling to represent the confidence intervals as a function of the error / uncertainty in the various reconstructions, perhaps through some Monte Carlo sampling? This would help to determine if shifts to positive correlations in Fogt reconstruction in 1955 or the proxy-based reconstructions in years 1100-1300 are different than zero when accounting for the uncertainty in the reconstructions. I understand this a goal of Figure 2, but you can also do this in the real-world sense in Figure 1.

2. I have trouble interpreting Figure 4, since it is based on the sign of the correlation, rather than the phases of either ENSO or SAM. This would mean the negative composite, for example, contains years of ENSO+ and SAM- (El Nino with SAM-) as well as ENSO- and SAM+ (La Nina with SAM+). One would expect this would lead to cancellation of many of the circulation features since the phase of the two modes are opposite, and indeed in Fig. 4 you do not see any SAM signatures in SLP over Antarctica, or really any ENSO signatures in temperature or SLP in the tropical Pacific. I suggest redoing these to have a more meaningful result, since previous work suggests high southern latitude ENSO teleconnections are stronger when there is a negative ENSO-SAM correlation. To do this, you can continue to use the correlation as a tool to select years, but then make sure to adjust the anomalies based on the phases of ENSO / SAM before compositing to be consistent and avoid cancellation. For example, you could multiply the years of SAM- and El Nino by negative 1 before adding these two the years with SAM+ and La Nina, to represent the circulation specifically during negative correlation events. I think this would be much more meaningful. The IPO signature may emerge more than the ENSO one since it is a persistent mode of variability, whereas ENSO and SAM change phase much more frequently.

3. Also, it is odd to have a focus on the Aleutian Low in Fig. 4 – why would this be influenced by ENSO and SAM? A more robust measure would be in the SH (where SAM has a direct influence), such as the Amundsen Sea Low, which is known to change in its magnitude based on ENSO / SAM relationships. Compositing Fig. 4 in the fashion described above should make for a clear connection with the ASL.

Minor suggestions:

1. Figure 1 – suggest clarifying that ENSO is based on the Nino 3.4 index, as some other studies have used the SOI and therefore had a positive correlation

2. The authors may wish to cite / incorporate a recent study which looked at stationarity in pressure relationships between the mid / high latitudes of the SH over the last century in models and reconstruction, also using perfect pseudoproxy model reconstructions (Clark and Fogt 2019):

Clark, L., and R. Fogt, 2019: Southern Hemisphere Pressure Relationships during the 20th Century—Implications for Climate Reconstructions and Model Evaluation. Geosciences, 9, 413, https://doi.org/10.3390/geosciences9100413.

---

## Referee Comment (RC2) · Anonymous Referee #2 · 4 Dec 2019

Dätwyler et al. study the correlation between ENSO and SAM using the CESM last millennium ensemble and two proxy-based reconstructions. They argue for a variable relationship, with a long-term mean anti-correlation of around -0.3. This is in agreement with previous work.

The paper is overall well-written, well-referenced and easy to follow. To me the strongest point is the pseudo-proxy validation effort of the ENSO and SAM records. Since these reconstructions were made by Dätwyler himself, this effort would perhaps have been more appropriately discussed in the original papers that presented these records. While the analyses appear to be technically valid, I am not sure whether

the work contributes meaningfully to our understanding of atmospheric dynamics (see comments below). The focus is purely on correlation analysis of various indices, which is a strong limitation of the study. A discussion of the potential and limitations of such an approach is needed. Also, including an understanding of the underlying climate dynamics would make for a more mean interesting paper.

Comments:

Since there are many different indices of ENSO and its teleconnections, as well as different indices of the SAM, it is not clear to me whether the presented correlation represents fundamental atmospheric dynamics or just an artifact of how the indices are defined. A different choice of indices may give a different result. For example, the ENSO teleconnection to the South Pacific is often associated with the PSA1 pattern. The SAM and PSA1 patterns are commonly defined as the first and second EOF pattern of SH extra-tropical geopotential height anomalies – in which definition they are orthogonal by construction and therefore have a correlation of r=0. Positive ENSO/PSA1 is associated with a weakened Amundsen Sea low via a Rossby wave train; all else being equal this will show up as a negative SAM phase in the definition of the authors (positive SLP anomaly at 65S), even without any SAM-specific dynamics being involved. I think the authors should check whether the correlation they observe is an artifact of the choice of indices, or something more fundamental. At least this important caveat should be discussed.

The long-term average correlation is weak (r around -0.3, or less than 10% of variance explained). Two timeseries that are weakly correlated will always have periods of stronger and weaker correlation. Since the extremes in this running-window correlation are not very obviously linked to any climate patterns (Fig. 4), could this pattern simply arise from the autocorrelation of both time series? Also, with only 31 years in the running window, I imagine anything below |r| = 0.35 or so may not exceed 95% confidence. Are these variations statistically significant?

[Figure]

The authors implicate the Aleutian low in periods where the SAM-ENSO relationship is anomalous. Is there a good dynamical explanation for this? I think of the Aleutian Low as the NH analog of the Amundsen Sea low. Given that ENSO teleconnection patterns have hemispheric symmetry [Seager et al., 2003], could this simply be a NH ENSO teleconnection? I find it hard to believe the Aleutian low is actually driving these SH high-latitude phenomena; more likely it is just a downstream indicator of certain anomalies in tropical Pacific convection.

There is little discussion of, or insight into, the atmospheric dynamics that may drive these observations. What does this correlation represent, and why is it important in the first place? Does it reflect an influence of ENSO on SAM (most likely, right?) or vice versa?

How is it possible that the correlation in 1800-2000 CE is stronger in noisy pseudo proxies (Fig. 3b) than in noise-free pseudo proxies? This is very counter-intuitive. Is the noise applied to the ENSO and SAM proxies correlated? In the supplement it is mentioned that several proxies are used in both reconstructions – could this produce such an unexpected result?

The authors only show the 31-yr running window correlations. However, both ENSO activity and the SAM have long-term (multi-centennial) variations also, that are not captured in such an analysis. What correlation do you get for different window lengths? What is the correlation of the full time series without any filtering?

The authors conclude (surprisingly) that the no-noise pseudo-proxies may be a better representation of the real world than the noisy pseudo proxies (Section 3.3). It seems this is based on the red and black lines overlapping in Figs. 3c and 3d. However, the reconstructions use multiple types of proxies – could it be that different proxies have different SNR levels? Also, the SAM red curve seems to overlap with the blue line better in the early part of the reconstruction – wouldn't that imply that SNR=0.5 is a good choice for the SAM reconstruction after all?

[Figure]

Minor comments:

L21 and elsewhere: What are "negative teleconnections"? I suppose you mean negative SAM-ENSO correlations. This is not the same. Please replace here and throughout the paper.

L33: "little is known about the interplay between them"... Elsewhere you cite a handful of papers on this topic, I think you should cite them here also.

L41: Negative correlation: Please describe in words what this means climatically. I assume it is a positive ENSO phase (El Nino) coinciding with a negative SAM phase (weak vortex, SH westerlies displaced northward).

L154-156: Why is the interval chosen asymmetrically at +3 to -2 stdev?

L169: correlation strength is in the eye of the beholder, but I would not call a correlation in the 0.2 to 0.3 range strong. That is only 5-10% of variance explained.

L170, L180 and throughout: "breakdown" seems an inappropriate word. In weakly correlated time series like these there will always be periods of stronger and weaker correlation. Nothing may have changed.

L189: observational data is confusing here. Do you mean proxy or instrumental data? Both are observational. Please clarify.

L204: I interpret the trend in Fig 3a is an artifact of the proxy reconstruction, given that the model relationship is steady (Fig. 2). Is that correct? Please state this more explicitly.

L207: why do the noisy proxies have stronger correlations than the no-noisy proxies?

L226: The binary choice between SNR=0.5 and perfect proxies is of course artificial. Can you estimate the optimal SNR value? Surely there must be some noise in the proxies (realistically a different SNR for each proxy type/archive).

L255: Can you please remind the reader how these are defined? What years are averaged?

L291: "teleconnections break apart or are particularly strong" should be: correlations are weak or particularly strong. You're not studying the teleconnection patterns directly, this is all extrapolation.

Figure 1: Can you show the correlation for longer time windows also? Which of these values are statistically significant? With only 31 years in the window, I imagine anything below r = 0.35 or so does not exceed 95% confidence. Last, note that DJF is when ENSO is strongest, but SAM feedbacks and lifetime are strongest in spring (SON).

Figure 4: You note that the model strongly underestimates the variability in the ENSO-SAM correlation, suggesting it may lack the relevant dynamics that drives this in the real world – maybe state this caveat when interpreting this figure. On the left y-axes labels, replace "teleconnection" with "correlation". How is statistical significance determined? Do you compare the anomalous years to the statistics of the full model run? It is surprising that the strongest SAT and SLP anomalies are not statistically significant.

Reference: Seager, R., N. Harnik, Y. Kushnir, W. Robinson, and J. Miller (2003), Mechanisms of Hemispherically Symmetric Climate Variability, J. Clim., 16(18), 2960-2978, doi: 10.1175/1520-0442(2003)016<2960:MOHSCV>2.0.CO;2.

---

## Author Comment (AC1) · 29 Jan 2020

Overview – the authors are examining relationships between austral summer ENSO and SAM indices from reconstructions and model simulations over the last millennia. As these relationships are important for understanding climate variations across the mid and high latitudes of the Southern Hemisphere, and no research to my knowledge has been done on these relationships this far back in time, the work is very important. The paper is extremely well written, concise, and easy to follow. I enjoyed it very much, and suggest it be published after the authors address my three main concerns, and

two very minor suggestions.

Major concerns: 1. Figure 1 – is there a way that uncertainty in the reconstructions (both the Fogt and the proxy-based) can be accounted for when calculating the correlation, and this can be shown as some sort of confidence interval around the correlations? While you can do this as a statistical test that the correlation is zero (95% confidence interval on the correlation magnitude itself), I think it would be more telling to represent the confidence intervals as a function of the error / uncertainty in the various reconstructions, perhaps through some Monte Carlo sampling? This would help to determine if shifts to positive correlations in Fogt reconstruction in 1955 or the proxy-based reconstructions in years 1100-1300 are different than zero when accounting for the uncertainty in the reconstructions. I understand this a goal of Figure 2, but you can also do this in the real-world sense in Figure 1.

Response 1: Yes. We calculated the significance in two different ways and shaded values exceeding the 95% confidence threshold with grey and cyan colour respectively. The cyan shading corresponds to significant individual 31-year window correlations (p<0.05), taking auto-correlation of lag 1 into account. The grey shading represents values that exceed the 95% confidence range obtained from the uncertainty in the reconstructions. The 95% confidence range was calculated from the running correlations of 1000 ensemble members of the SAM and ENSO reconstructions. For the SAM reconstruction the data can be downloaded from https://www.ncdc.noaa.gov/paleo-search/study/23130 and for ENSO, an ensemble of 1000 reconstructions was generated by adding noise to the reconstruction with the same AR1 coefficient as the reconstruction and variance equal to the square of two times the augmented standard deviation of the residuals between the reconstruction and target ENSO index over the reference period 1930-1990 (SDres.aug, formula as in Dätwyler et al. 2019).

2. I have trouble interpreting Figure 4, since it is based on the sign of the correlation, rather than the phases of either ENSO or SAM. This would mean the negative composite, for example, contains years of ENSO+ and SAM- (El Nino with SAM-) as

well as ENSO- and SAM+ (La Nina with SAM+). One would expect this would lead to cancellation of many of the circulation features since the phase of the two modes are opposite, and indeed in Fig. 4 you do not see any SAM signatures in SLP over Antarctica, or really any ENSO signatures in temperature or SLP in the tropical Pacific. I suggest redoing these to have a more meaningful result, since previous work suggests high southern latitude ENSO teleconnections are stronger when there is a negative ENSO-SAM correlation. To do this, you can continue to use the correlation as a tool to select years, but then make sure to adjust the anomalies based on the phases of ENSO / SAM before compositing to be consistent and avoid cancellation. For example, you could multiply the years of SAM- and El Nino by negative 1 before adding these two the years with SAM+ and La Nina, to represent the circulation specifically during negative correlation events. I think this would be much more meaningful. The IPO signature may emerge more than the ENSO one since it is a persistent mode of variability, whereas ENSO and SAM change phase much more frequently.

Response 2: Thank you for the suggestion. Note that a negative correlation can not only arise during ENSO+ with SAM- and ENSO- with SAM+, but may also occur during ENSO+ with SAM+ and ENSO- with SAM- when both vary opposed to each other. But we agree that our approach may lead to the cancellation of circulation features and will recalculate the patterns in Fig. 4 taking the phases of ENSO and SAM into account and test the suggested multiplication of temperature/SLP patterns by negative 1 of either years with SAM+ and La Niña or SAM- and El Niño.

3. Also, it is odd to have a focus on the Aleutian Low in Fig. 4 – why would this be influenced by ENSO and SAM? A more robust measure would be in the SH (where SAM has a direct influence), such as the Amundsen Sea Low, which is known to change in its magnitude based on ENSO / SAM relationships. Compositing Fig. 4 in the fashion described above should make for a clear connection with the ASL.

Response 3: Referee #2 notes that the Aleutian low pattern we see in Fig. 4 is possibly only a teleconnection related to ENSO. We agree with this. At the current state of

knowledge, we cannot provide a sound dynamical explanation how the Aleutian low stands in relation to ENSO-SAM correlations. Therefore, we decided to restrict our analysis of spatial patterns to the SH only, where a direct link to both ENSO and SAM is more obvious. The text in the manuscript at all relevant places will be changed accordingly.

Minor suggestions:

1. Figure 1 – suggest clarifying that ENSO is based on the Nino 3.4 index, as some other studies have used the SOI and therefore had a positive correlation

Response 4: Ok, done.

2. The authors may wish to cite / incorporate a recent study which looked at stationarity in pressure relationships between the mid / high latitudes of the SH over the last century in models and reconstruction, also using perfect pseudoproxy model reconstructions (Clark and Fogt 2019): Clark, L., and R. Fogt, 2019: Southern Hemisphere Pressure Relationships during the 20th CenturyâËŸAËĞ TImplications for Climate Reconstructions and Model Evaluation. Geosciences, 9, 413, https://doi.org/10.3390/geosciences9100413.

Response 5: Ok.

**Fig. 1.** Updated Fig. 1. Added shading of significant correlations.

---

## Author Comment (AC2) · 29 Jan 2020

Dätwyler et al. study the correlation between ENSO and SAM using the CESM last millennium ensemble and two proxy-based reconstructions. They argue for a variable relationship, with a long-term mean anti-correlation of around -0.3. This is in agreement with previous work.

The paper is overall well-written, well-referenced and easy to follow. To me the strongest point is the pseudo-proxy validation effort of the ENSO and SAM records.

Since these reconstructions were made by Dätwyler himself, this effort would perhaps have been more appropriately discussed in the original papers that presented these records. While the analyses appear to be technically valid, I am not sure whether the work contributes meaningfully to our understanding of atmospheric dynamics (see comments below). The focus is purely on correlation analysis of various indices, which is a strong limitation of the study. A discussion of the potential and limitations of such an approach is needed. Also, including an understanding of the underlying climate dynamics would make for a more mean interesting paper.

Response 6: We agree that the inclusion of dynamical explanations of the observed phenomena would be very interesting. At the current stage of knowledge, however, we are yet far away from being able to provide a sound dynamical interpretation of the results presented in this manuscript. Due to ENSO's and SAM's far-reaching teleconnections, there are many oceanic and atmospheric process involved that may play a role and/or be affected/modulated by these two large scale modes of climate variability. Including an extended analysis and understanding the underlying dynamical processes in the climate system is beyond the scope of this paper. Furthermore, such interpretations are outside the expertise of the author team and subject to further research. This study should be seen as a first, purely statistical approach to shed light on the relationship between ENSO and SAM over the last millennium and spark hypotheses for possible dynamic linkages that need to be explored and tested. At the end of the introduction we expanded the text by a statement that more clearly defines the scope of this paper: "Whereas an assessment and interpretation of underlying dynamical processes in the climate system are beyond the scope of this study, this work provides some first insights in spatial patterns of climate. Spatial patterns during periods of particularly strong negative or reversed positive ENSO-SAM relationships are analysed in the model world to identify potential driving factors of SH teleconnection changes."

Comments:

Since there are many different indices of ENSO and its teleconnections, as well as

different indices of the SAM, it is not clear to me whether the presented correlation represents fundamental atmospheric dynamics or just an artifact of how the indices are defined. A different choice of indices may give a different result. For example, the ENSO teleconnection to the South Pacific is often associated with the PSA1 pattern. The SAM and PSA1 patterns are commonly defined as the first and second EOF pattern of SH extra-tropical geopotential height anomalies – in which definition they are orthogonal by construction and therefore have a correlation of r=0. Positive ENSO/PSA1 is associated with a weakened Amundsen Sea low via a Rossby wave train; all else being equal this will show up as a negative SAM phase in the definition of the authors (positive SLP anomaly at 65S), even without any SAM-specific dynamics being involved. I think the authors should check whether the correlation they observe is an artifact of the choice of indices, or something more fundamental. At least this important caveat should be discussed.

Response 7: We agree that a specific choice of indices and their definition can lead to artefacts in the reconstructions and observed correlations. We checked (see figure below) whether a different choice of indices would lead to fundamentally different results. The running correlations between SAM and ENSO reconstructions that are based on different indices are calculated over a period of relatively good proxy coverage (1800-present) to minimise the chance that the signals are confounded too heavily by proxy noise. For SAM we use reconstructions of the Fogt and Marshall indices and for ENSO reconstructions of the Niño3.4 (based on two different data sets), Niño1+2 and SOI (multiplied by -1). In comparison to the index reconstruction used in this manuscript, we see the most pronounced differences when comparing them to the Niño1+2 index reconstruction. This difference likely arises by the fact that the Niño1+2 index represents a more regional coastal flavour of ENSO. All correlations confirm the significant negative correlations between the two modes of climate variability, except for one case that was not significant (SAM Fogt vs. Niño1+2). All other cases even yield a stronger significant negative correlation (from -0.28 to -0.47, opposed to -0.25 for the index reconstructions used in this manuscript). Hence, we are confident to say

that the observed negative correlations do not arise from an artefact of a specific index or choice thereof. In fact, the comparison of the correlations based on the different indices suggests that the apparent decline of the correlations between ca. 1950-1980 seen in our data (Fig.1), may be an artefact of the choice of the index (SAM Fogt). Results based on the SAM Marshall suggest that correlations were negative during the entire 20th century, more strongly confirming the general negative correlation pattern seen in the model data. We will include a statement on the outcome of this comparison for the choice of different indices in the main text and add the below figure to the supplementary material.

The long-term average correlation is weak (r around -0.3, or less than 10% of variance explained). Two timeseries that are weakly correlated will always have periods of stronger and weaker correlation. Since the extremes in this running-window correlation are not very obviously linked to any climate patterns (Fig. 4), could this pattern simply arise from the autocorrelation of both time series? Also, with only 31 years in the running window, I imagine anything below |r| = 0.35 or so may not exceed 95% confidence. Are these variations statistically significant?

Response 8: See also "Response 1" (response to Referee#1) and "Response 12". When calculating the significance, lag 1 auto-correlation is taken into account. As the reviewer writes, having only 31 years in the correlation window leads to the fact that not too many periods exceed 95% confidence. However, the model results (Fig. 2) clearly suggest that, from the perspective of the dynamics in the climate models, the correlation is significant basically over the entire millennium (with individual model runs also showing large periods of non-significant correlations as in the real-world case).

The authors implicate the Aleutian low in periods where the SAM-ENSO relationship is anomalous. Is there a good dynamical explanation for this? I think of the Aleutian Low as the NH analog of the Amundsen Sea low. Given that ENSO teleconnection patterns have hemispheric symmetry [Seager et al., 2003], could this simply be a NH ENSO teleconnection? I find it hard to believe the Aleutian low is actually driving these

SH high-latitude phenomena; more likely it is just a downstream indicator of certain anomalies in tropical Pacific convection.

Response 9: We agree that the link to the Aleutian low is possibly only a teleconnection related to ENSO and at the current state of knowledge, we cannot provide a sound dynamical explanation how the Aleutian low stands in relation to ENSO-SAM correlations. Therefore, we decided to restrict our analysis of spatial patterns to the SH only, where a direct link to both ENSO and SAM is more obvious. The text in the manuscript at all relevant places will be changed accordingly.

There is little discussion of, or insight into, the atmospheric dynamics that may drive these observations. What does this correlation represent, and why is it important in the first place? Does it reflect an influence of ENSO on SAM (most likely, right?) or vice versa?

Response 10: See also "Response 6" and "Response 9". Yes, the literature generally speaks for a driving influence of ENSO on SAM (e.g., Ribera and Mann, 2003; Carvalho et al., 2005, Fogt and Bromwich, 2006; L'Heureux and Thompson, 2006; Cai et al., 2010; Gong et al., 2010; Yu et al., 2015; Kim et al., 2017). E.g., Carvalho et al. (2005) propose that low-frequency variability in central Pacific SSTs modulating tropical convection patterns lead to variations in the SAM. Yu et al. (2015) describe an influence of ENSO on SAM through both "an eddy-mean flow interaction mechanism and a stratospheric pathway mechanism". Fogt et al. (2011) and Kim et al. (2017) find that influence of ENSO on SAM is linked to eddy momentum flux and associated wave propagation. Since both modes of climate variability play an important role for SH climate, a better long-term understanding of the ENSO-SAM teleconnections will lead to more precise predictions of present and future climate patterns across the SH. However, an interpretation of underlying dynamical processes will be subject to future research and is beyond the scope of this study.

How is it possible that the correlation in 1800-2000 CE is stronger in noisy pseudo

proxies (Fig. 3b) than in noise-free pseudo proxies? This is very counter-intuitive. Is the noise applied to the ENSO and SAM proxies correlated? In the supplement it is mentioned that several proxies are used in both reconstructions – could this produce such an unexpected result?

Response 11: There is an overlap of only three records used in both reconstructions. We re-calculated the ENSO reconstruction removing these three proxy records. The correlation between the original and re-calculated reconstruction is r = 0.994 (over the period 1800-present). Hence, we argue that the overlap of proxy records does not influence our conclusions and has not lead to unexpected results. Although the correlation between the noisy pseudo-proxy reconstruction is slightly stronger over 1800-2000, the difference is not significant (uncertainties overlap). As described in the methods section and supplement, the perfect pseudo-proxies were all only allocated to a single climate variable (either temperature or precipitation). In the case of noise pseudo-proxies, proxy system models were used for tree-ring widths and $\delta$18O coral records. These PSM include both temperature and precipitation or SST and salinity, respectively. We therefore hypothesise, that this has led to an additional signal in the noisy pseudo-proxies, especially for the coral pseudo-proxies that, like the real-world proxies, drop out significantly prior to 1800. We think this effect is what is leading to the observed slightly stronger negative correlation over 1800-2000.

The authors only show the 31-yr running window correlations. However, both ENSO activity and the SAM have long-term (multi-centennial) variations also, that are not captured in such an analysis. What correlation do you get for different window lengths? What is the correlation of the full time series without any filtering?

Response 12: We will include a plot in the supplementary material that shows correlations for different window widths (see figure below). The correlation over the full time series (using the real-world proxy-based reconstructions) without any filtering (taking lag-1 auto-correlation into account) is -0.07 (not significant). Over 1200-1990 it's -0.09 (significant, p<0.05), over 1400-1990 it's -0.16 (significant, p<0.01), over 1600-1990 it's

-0.17 (significant, p<0.01) and over 1800-1990 it's -0.25 (significant, p<0.01). I.e., we get significant negative correlations as far back as to 1200. There are only few proxy records that contribute to the reconstructions in the early most part which leads to a significant amount of noise. The decrease in correlation strength can also be seen in the pseudo-proxy experiments and also suggests that it is caused by proxy-inherent noise.

The authors conclude (surprisingly) that the no-noise pseudo-proxies may be a better representation of the real world than the noisy pseudo proxies (Section 3.3). It seems this is based on the red and black lines overlapping in Figs. 3c and 3d. However, the reconstructions use multiple types of proxies – could it be that different proxies have different SNR levels? Also, the SAM red curve seems to overlap with the blue line better in the early part of the reconstruction – wouldn't that imply that SNR=0.5 is a good choice for the SAM reconstruction after all?

Response 13: Yes, the conclusion that the perfect pseudo-proxies may be a better representation of the real world than the noisy pseudo proxies is based on Figs. 3c/3d. For ENSO, the correlation between the instrumental index and the real-world reconstruction (red line) is much closer to the correlation between the model ENSO and perfect pseudo-proxy reconstructions (black line) than between the model ENSO and noisy pseudo-proxy reconstructions (blue line). The same holds true in the case of SAM when using the Marshall SAM index and for the more recent halve when using the Fogt SAM index. Indeed, the correlations drop down to the blue line over the earlier half for the correlation between the Fogt SAM index and the real-world SAM reconstruction. But this happens about just around the end of the 1950s back to when reliable observational data for the SAM exist. Hence, given the good match between the red and black line for ENSO, the green and black line for SAM (Marshall) and the red and black line over the recent halve for SAM (Fogt) where the instrumental data are most reliable, we conclude that a SNR of 0.5 may be too conservative and the no-noise proxies may come closer to the real-world. Concerning different SNR levels for different proxy types:

For three-ring width and $\delta$18O coral records proxy-system-models are used to generate the pseudo-proxy records. The resulting pseudo-proxies yielded a SNR of about 0.5. Furthermore, based on the relationship between proxy records and instrumental data, a SNR ratio in the range of 0.5 is generally considered to be realistic (e.g., 0.45 in Wang et al. 2014, Steiger and Hakim 2016). For reconstructions of large-scale or hemispheric phenomena, a higher SNR between 0.5 and 1 is proposed (Neukom et al. 2018) since in models (and specifically the CESM model used herein), the correlation between the large-scale target and local climate is too low compared to observations. Model-based pseudoproxies are therefore noisier than real world proxies with regards to proxy-target correlations. In summary, to the current state of knowledge, an estimate of proxy noise with an SNR of 0.5 seems realistic, with a tendency to lie on the conservative side (which also seems to be confirmed by the results described above).

Minor comments:

L21 and elsewhere: What are "negative teleconnections"? I suppose you mean negative SAM-ENSO correlations. This is not the same. Please replace here and throughout the paper.

Response 14: Ok, done.

L33: "little is known about the interplay between them"... Elsewhere you cite a handful of papers on this topic, I think you should cite them here also.

Response 15: Ok, done.

L41: Negative correlation: Please describe in words what this means climatically. I assume it is a positive ENSO phase (El Nino) coinciding with a negative SAM phase (weak vortex, SH westerlies displaced northward).

Response 16: Yes, but also vice versa. We extended the text to describe this as suggested by the reviewer. It now reads: "A negative correlation arises when a change in ENSO towards more El Niño-like (La Niña-like) conditions coincides with a shift in

the SAM towards a more negative (positive) phase, which is associated with a shift of the storm tracks towards the Equator (South Pole) resulting in weaker (stronger) circumpolar westerly winds."

L154-156: Why is the interval chosen asymmetrically at +3 to -2 stdev?

Response 17: The average correlation is negative (-0.3). Choosing +3 and -2 standard deviations leads to thresholds of +0.26 and -0.67 and 53 and 281 years that exceed these thresholds, respectively. There is a trade-off between the amount of years that go into the analysis and the expected strength of the associated signal in the climate patterns emerging from the years of strong negative or positive correlations. Whereas for negative correlations a threshold of -0.67 together with 281 years in the sample seemed reasonable to us, a standard deviation of +2 from the mean would lead to a threshold correlation of +0.07 (and 434 years). Because in our opinion a positive correlation of only +0.07 as cut-off value was too close to zero, we decided to choose +3 instead of +2 standard deviations from the mean leading to the above stated numbers (+0.26 and 53 years). We are aware of the fact that such decisions are, to some extent, subjective. Therefore, different values have been tested and showed the robustness of the results with respect to these choices (Section S5 in the supplementary material).

L169: correlation strength is in the eye of the beholder, but I would not call a correlation in the 0.2 to 0.3 range strong. That is only 5-10% of variance explained.

Response 18: Rephrased from "Particularly during the period of high data quality back to the 1970s, the relationship is strongly negative." to "Particularly during the period of high data quality back to the 1970s, the indices are significantly negatively correlated with a value of -0.38 (1980-2019)."

L170, L180 and throughout: "breakdown" seems an inappropriate word. In weakly correlated time series like these there will always be periods of stronger and weaker correlation. Nothing may have changed.

Response 19: Replaced. The text has changed as follows: - "In the longer and less certain instrumental dataset (Fogt SAM index), a breakdown of the SAM-ENSO relationship is visible around 1955." now reads "In the longer and less certain instrumental dataset (Fogt SAM index), a change in the SAM-ENSO relationship is visible around 1955." - "To test whether the breakdowns in the reconstructed ENSO-SAM correlations back in time and the lack of response to external forcing are realistic features or an artefact of decreasing proxy data availability and quality, we now compare the results with correlations from model simulations." is changed to "To test whether the fluctuations in the reconstructed ENSO-SAM correlations back in time and the lack of response to external forcing are realistic features or an artefact of decreasing proxy data availability and quality, we now compare the results with correlations from model simulations." - "In addition to this consistent average negative response, the individual 13 running correlations for each simulation also show periods with breakdowns in the negative correlation and intermittent changes to positive relationships (Fig. S3)." now reads "In addition to this consistent average negative response, the individual 13 running correlations for each simulation also show periods with strong fluctuations in the negative correlation and intermittent changes to positive relationships (Fig. S3)." - "In case of noisy pseudo-proxies for the ENSO, the strength of the correlations decreases significantly prior to 1800 when they remain positive but on a lower level. In the case of SAM, the noisy pseudo-proxies do not result in an equally strong decrease in the strength of the running correlations. Correlations are about 0.2 lower for the noisy PPE throughout most of the reconstruction period; only around 1300 they start to break down to levels around 0.25 between 1000-1200." is rephrased to "In the case of noisy pseudo-proxies for the ENSO, the strength of the correlations decreases significantly prior to 1800 when they remain positive but on a lower level. In the case of SAM, the noisy pseudo-proxies do not result in an equally strong decrease in the strength of the running correlations. Correlations are about 0.2 lower for the noisy PPE throughout most of the reconstruction period; only around 1300 they start to move to levels around 0.25 between 1000-1200." - "The rapid decrease in the strength of the signal in the

noisy PPE prior to 1800 due to proxy noise may be over-pessimistic and the gentler decrease in the perfect PPE a more realistic representation of the real-world situation." is changed to "The rapid decrease in the strength of the signal in the noisy PPE prior to 1800 due to proxy noise may be over-pessimistic and the gentler decrease in the perfect PPE a more realistic representation of the real-world situation."

L189: observational data is confusing here. Do you mean proxy or instrumental data? Both are observational. Please clarify.

Response 20: Clarified. "Similar to the observational data, . . ." now reads "Similar to the real-world proxy-based reconstructions, . . .".

L204: I interpret the trend in Fig 3a is an artifact of the proxy reconstruction, given that the model relationship is steady (Fig. 2). Is that correct? Please state this more explicitly.

Response 21: We wouldn't call this trend an artefact of the proxy reconstruction. Rather, this is an expected phenomenon that relates to the number of proxy records contributing to the reconstruction over time. The fewer proxies that contribute to the reconstruction, the higher the noise level in the reconstruction. In contrast to Fig. 2, where the amount of noise does not change over time, the number of records contributing to the reconstruction in Fig. 3 decreases over time and therefore, it is not unexpected that the correlations decrease due to the increased amount of noise in the reconstructions back in time. We adapted the text from ". . . we can observe that the running correlations stay negative over the whole millennium with a slight trend towards more negative correlations towards present (Fig. 3a)." to ". . . we can observe that the running correlations stay negative over the whole millennium with a slight trend (originating from the reduced number of pseudo-proxy records contributing to the reconstructions back in time) towards more negative correlations towards present (Fig. 3a)."

L207: why do the noisy proxies have stronger correlations than the no-noisy proxies?

Response 22: "See Response 11".

L226: The binary choice between SNR=0.5 and perfect proxies is of course artificial. Can you estimate the optimal SNR value? Surely there must be some noise in the proxies (realistically a different SNR for each proxy type/archive).

Response 23: See "Response 13".

L255: Can you please remind the reader how these are defined? What years are averaged?

Response 24: Ok. We expanded the text, which is now reading "To identify such patterns, we analyse similar anomalies in the model data by averaging all years where the correlations are above or below the -2/+3 standard deviation threshold, as described in the methods section".

L291: "teleconnections break apart or are particularly strong" should be: correlations are weak or particularly strong. You're not studying the teleconnection patterns directly, this is all extrapolation.

Response 25: Changed as suggested.

Figure 1: Can you show the correlation for longer time windows also? Which of these values are statistically significant? With only 31 years in the window, I imagine anything below r = 0.35 or so does not exceed 95% confidence. Last, note that DJF is when ENSO is strongest, but SAM feedbacks and lifetime are strongest in spring (SON).

Response 26: See "Response 1" (response to Referee#1) and "Response 12".

Figure 4: You note that the model strongly underestimates the variability in the ENSOSAM correlation, suggesting it may lack the relevant dynamics that drives this in the real world – maybe state this caveat when interpreting this figure. On the left y-axes labels, replace "teleconnection" with "correlation". How is statistical significance determined? Do you compare the anomalous years to the statistics of the full model run? It

is surprising that the strongest SAT and SLP anomalies are not statistically significant.

Response 27: Ok. Replaced "teleconnections" with "correlations". As described in the methods section, significance is determined by re-calculating the composites 1000 times using random years (same number of years as used for the composites that are based on ENSO-SAM correlations). Therefore, regions with stronger SST and SLP variability can also yield stronger anomalies in the random composites, thus leading to larger confidence intervals for the null compared to other regions. Thus, significance is relative to the variability at each grid-cell and may be easier to reach in regions with lower variance.

Reference: Seager, R., N. Harnik, Y. Kushnir, W. Robinson, and J. Miller (2003), Mechanisms of Hemispherically Symmetric Climate Variability, J. Clim., 16(18), 2960-2978, doi: 10.1175/1520-0442(2003)016<2960:MOHSCV>2.0.CO;2.

**31-year running correlations between reconstructions of different SAM and ENSO indices**

| | |
|---|---|
| — SAM Fogt vs. Niño3.4 (ERSSTv4) \| -0.25 | - - SAM Marshall vs. Niño3.4 (ERSSTv4) \| -0.41 |
| — SAM Fogt vs. SOI \| -0.36 | - - SAM Marshall vs. SOI \| -0.46 |
| — SAM Fogt vs. Niño3.4 (Kaplan) \| -0.37 | - - SAM Marshall vs. Niño3.4 (Kaplan) \| -0.47 |
| — SAM Fogt vs. Niño1+2 (ERSSTv4) \| -0.19 | - - SAM Marshall vs. Niño1+2 (ERSSTv4) \| -0.28 |

**Fig. 1.** 31-year running correlations between reconstructions of different SAM and ENSO indices. The numbers in the legend refer to the correlation over the whole period.

[Figure]

**Fig. 2.** Running correlations between ENSO and SAM DJF reconstructions for different window widths.